# Suppression of an Invasive Native Plant Species by Designed Grassland Communities

**DOI:** 10.3390/plants10040775

**Published:** 2021-04-15

**Authors:** Kathrin Möhrle, Hugo E. Reyes-Aldana, Johannes Kollmann, Leonardo H. Teixeira

**Affiliations:** 1Restoration Ecology, School of Life Sciences, Technical University of Munich, 85354 Freising, Germany; jkollmann@wzw.tum.de (J.K.); leonardo.teixeira@tum.de (L.H.T.); 2Bavarian State Research Centre for Agriculture, Institute for Crop Science and Plant Breeding, 85354 Freising, Germany; 3Department of River Ecology, Helmholtz Center for Environmental Research—UFZ, 39114 Magdeburg, Germany; hugorealdana.biot@gmail.com; 4Norwegian Institute of Bioeconomy Research, P.O. Box 115, 1431 Ås, Norway

**Keywords:** design of seed mixtures, grassland establishment, native invasive species, phylogenetic diversity, trait-based restoration

## Abstract

Grassland biodiversity is declining due to climatic change, land-use intensification, and establishment of invasive plant species. Excluding or suppressing invasive species is a challenge for grassland management. An example is *Jacobaea aquatica*, an invasive native plant in wet grasslands of Central Europe, that is causing problems to farmers by being poisonous, overabundant, and fast spreading. This study aimed at testing designed grassland communities in a greenhouse experiment, to determine key drivers of initial *J. aquatica* suppression, thus dismissing the use of pesticides. We used two base communities (mesic and wet grasslands) with three plant traits (plant height, leaf area, seed mass), that were constrained and diversified based on the invader traits. Native biomass, community-weighted mean trait values, and phylogenetic diversity (PD) were used as explanatory variables to understand variation in invasive biomass. The diversified traits leaf area and seed mass, PD, and native biomass significantly affected the invader. High native biomass permanently suppressed the invader, while functional traits needed time to develop effects; PD effects were significant at the beginning of the experiment but disappeared over time. Due to complexity and temporal effects, community weighted mean traits proved to be moderately successful for increasing invasion resistance of designed grassland communities.

## 1. Introduction

Grasslands are dramatically impacted by land-use change at various scales [1,2], resulting in altered ecosystems with increased sensitivity to climatic change [3]. Changing environmental conditions create opportunities for the establishment of alien plant species, but can also cause overabundance of native plants. Some of these species are undesirable because they have negative effects on nutrient cycles and fodder quality [4,5], and cause local extinction of native biodiversity due to competitive exclusion or niche displacement [6]. The mechanisms of alien plant invasions and their management have received much attention, while native invaders can have similar effects but are less well understood [7]. While environmental fluctuation might favor invaders with the capacity for rapidly occupying recently created empty niches [8], or quickly adapting to changed site conditions [9], biotic filtering will select for few winners among invaders and resident species of a recipient community [10]. In fact, the consideration of such biotic mechanisms during the design of grassland communities has become common practice. In crop farming systems, natural prevention of weeds can be achieved by the use of highly competitive forb and legume mixtures [11], while for grassland farming, low-diversity grass mixtures are preferred, mostly due to higher costs of diverse forb and legume mixtures [12]. However, as the widespread use of herbicides is harmful to the environment, attempts at controlling invasive plants in managed grasslands should focus on using community-based mechanisms, e.g., specifically designed seed mixtures.

Considering this community-based approach, increasing diversity aspects of designed grasslands may result in greater invasion resistance due to the rapid pre-emption of ecological niches, thus inhibiting the establishment of invasive plants via limitation of shared resources [13]. Furthermore, invasive plants do not have to be alien, as they can become overabundant in their home communities, thus presenting fast spreading rates and potentially negatively affecting species composition [14]. However, due to having most of the ecological niches fully occupied, home communities offer limited niche space to potential invaders [15]. In turn, niches and species interactions are distributed differently in non-home communities, offering additional niches to invaders, if they can match the competitive conditions [16]. The competitive effects of such invasive native species is, however, not higher than the one of other native species, but they are more tolerant and less impacted by competition with co-occurring species in non-home communities [17]. Thus, we expect non-home species to express less competitive interactions towards the invader, and the home community to have stronger competition, therefore hindering the growth of native invasive plants. Still, the dominant mechanisms depicting community assembly in home and non-home communities need more investigation to be applied as a tool for increasing invasion resistance.

Furthermore, there is the effect that a given community may exert via its biomass. Because the amount of biomass produced by the recipient community can indicate strong competition towards invaders, mostly due to the correlation between high productivity and resource capture [18,19], one could simply increase the proportion of highly productive species when designing grassland communities. Other efficient means for the design of grasslands are the manipulation of phylogenetic diversity (PD) and trait similarity between resident and invasive species. In fact, PD correlates with trait similarity, meaning that closely related species show similar trait values [20], thus strongly competing for similar resources [21]. Recent evidence shows that PD between plants of the recipient communities and the potential invaders can be manipulated as a restoration tool to reduce invasion impacts [22]. In turn, the direct manipulation of plant traits for the design of seed mixtures occurs independently of phylogenetic relations. One could, for example, focus on specific traits related to plant strategies [23,24,25] and, therefore, increase the possibility of having a strong specific competition effect towards the invader. Moreover, designing plant communities through their resource-use traits might increase resistance to invasion or disturbance [21], since invasive plants have traits mostly related to high levels of resource acquisition, consumption [26], and growth [27,28]. For example, plant height indicates competition, because it relates to increased aboveground biomass and, therefore, more access to light and other resources [29,30]. Increased leaf area is mostly related to resource acquisition through photosynthetic activity and, therefore, to higher growth rates [31]. In addition, the production of large amounts of small seeds with considerably high germination and dispersal probabilities also influences invasion success [32]. These factors, if considered for the design of a grassland community, should also affect its capacity to resist invaders.

In this study, we tested the resistance of experimental grasslands to *Jacobaea aquatica* (synonym for *Senecio aquaticus*, Hill, Asteraceae)*,* a native poisonous herb recently becoming overabundant in wet grasslands of Central Europe [33]. Here, *Jacobaea aquatica* has become a major invasive native plant [33,34]. This short-lived perennial contains pyrrolizidine alkaloids that are resistant to drying or ensiling, and therefore persistent in the fodder. This poses a threat to livestock and humans through transferring of the poisonous components within the food chain [35]. Establishment is most common in disturbed moist to wet grasslands with altered management, where the species responds to gaps with fast germination and high growth rates [32]. The plant forms rosettes, and large amounts of wind-dispersed seeds are produced in its second year [36]. It can flower already at low height and shortly after being cut [37,38], but is sensitive to shading, as observed in low-intensity or abandoned grasslands [39] [M.T. Krieger, unpublished results]. Although being native to wet grasslands, *J. aquatica* is considered in this study as ‘invasive species’ because of its population dynamics with rapid spread, overabundance, and potential impacts on farms [14].

We aimed at designing trait-based grassland communities with characteristics to suppress invasion during initial stages of community assembly. This allows for testing an alternative method to control plant invasion without the use of pesticides. While abiotic processes are considered to dominate during early successional stages, at later stages biotic processes become more important [40]. Thus, the following hypotheses were tested in a long-term greenhouse experiment on establishment of designed grassland communities sown with the invasive native *J. aquatica*: (i) mesic (non-home) grassland communities show less suppression of the invasive species than wet (home) communities; (ii) suppression increases with productivity (measured by native biomass) of the recipient communities; (iii) increasing the similarity of key plant traits (plant height, leaf area, seed mass) of the communities with the same traits of the invader increases suppression; (iv) higher phylogenetic diversity of the communities should lead to more suppression; and (v) suppression of the invasive species increases with time since community establishment. The resulting knowledge should be used for designing and testing further communities under field and long-term conditions, because restoration initiatives still struggle to establish native communities that are invasion-resistant [21].

## 2. Results

### 2.1. Invasive Biomass across Grassland Mixtures

Germination of all grassland species was around 65% (SD ± 21.8%) based on germination tests and published data (Appendix A). Plant cover was >80% in all experimental trays and across the three experimental periods. Average native biomass (per week) was 2.25 ± 0.39 g 1.5 dm^−2^ (mean ± SE) in the reference LfL mixture, while it was 1.33 ± 0.15 g 1.5 dm^−2^ for mesic native communities, and 1.33 ± 0.20 g 1.5 dm^−2^ for wet native communities. Biomass in monocultures of the native invasive *J. aquatica* was 0.45 ± 0.10 g 1.5 dm^−2^ (Appendix A), and regrowth of the native invader occurred after all cutting events.

Testing the effects of community type and manipulated plant traits showed that only diversified traits significantly affected *J. aquatica* biomass (χ^2^ = 24.1, df = 2, *p* < 0.0001), while community type and constrained traits had no effects (Appendix A). Moreover, there was a significant interaction between diversified traits and community type, most likely due to the variability of plant performance in different community types (χ^2^ = 14.7, df = 2, *p* = 0.0006; Appendix A). Biomass of *J. aquatica* varied significantly among grassland mixtures (χ^2^ = 39.3, df = 11, *p* < 0.001; Figure 1).

Maximum *J. aquatica* suppression compared to the reference mixture (LfL) was achieved by the M-Ph*Sm* mixture (constraining plant height and diversifying seed mass) with 89%, while the minimal suppression was observed in the grassland mixture M-Sm*La* (constraining seed mass and diversifying leaf area) with 64% (see Appendix A for all comparisons). Interestingly, both grassland mixtures, i.e., the one presenting the strongest and the one with weakest suppression, belonged to mesic grasslands. This indicates higher variation in the suppression of *J. aquatica* when invading mesic (non-home) grasslands. Additionally, invasive biomass was lowest in the LfL mixture and highest in the *J. aquatica* monoculture. There was no statistical difference in the suppressive effect of mesic and wet communities when analyzed globally, i.e., including the averaged biomass of *J. aquatica* across all grassland mixtures for each specific community type (χ^2^ = 0.03, df = 1, *p* = 0.86; Appendix A), while the wet communities showed higher suppression of *J. aquatica* in period 3 (Appendix A). Invader biomass hardly varied across the six wet communities, while marked differences were observed among mesic communities (Figure 1).

### 2.2. Factors Explaining Invasive Biomass

When checking which particular aspects in the different grassland mixtures could explain the suppression of the native invader, we observed significant effects for four of the five explanatory variables included in the analysis (Table 1). In fact, native biomass was the most important control of *J. aquatica* biomass (χ^2^ = 16.0, df = 1, *p* < 0.0001; Figure 2a). Such strong suppression of native (non-invasive) biomass indicates a direct competition for resources between natives and invader occurring in grassland mixtures composed by highly productive species. The CWM value of seed mass also exerted a significant influence on *J. aquatica* biomass (χ^2^ = 11.0, df = 1, *p* = 0.0009; Figure 2b). However, contrarily to the effects observed for native biomass, increasing seed mass values of grassland mixtures resulted in a positive effect on invasive biomass (Figure 2b). Similarly, increasing phylogenetic diversity of the grassland mixtures also had a significantly positive effect on invasive biomass (χ^2^ = 5.35, df = 1, *p* = 0.021; Figure 2d), whereas increased CWM values for leaf area resulted in significant suppression (χ^2^ = 4.21, df = 1, *p* = 0.040; Figure 2c). Finally, no effects could be observed on *J. aquatica* biomass from the manipulation of CWM plant height values in our grassland mixtures (χ^2^ = 0.27, df = 1, *p* = 0.604; Table 1).

The effects of community traits on the suppression of *J. aquatica* varied with time since start of the experiment (Figure 2). Native community biomass negatively affected *J. aquatica* biomass only after the first cutting event (Figure 2a). Contrarily, phylogenetic diversity produced a positive effect on *J. aquatica* biomass during the first experimental period, i.e., before the first cutting (Figure 2d). This might indicate that the phylogenetic relations among resident species of the recipient communities and the native invader play an important role in the early stages of community assembly [41]. CWM values for seed mass and leaf area were correlated with *J. aquatica* biomass in opposing directions, but significant effects were detected only in the third period, i.e., after two cuttings (Figure 2b,c). These late effects of the manipulated functional traits reveal that the role played by functional diversity in affecting the invasion resistance of grassland communities might increase with community development [42].

## 3. Discussion

The study aim was to evaluate the effects of designed communities consisting of different forb and legume species common to mesic or wet grasslands, on suppressing an invasive native plant during early stages of community assembly. The results show that all experimental grassland mixtures suppressed *J. aquatica*, confirming that revegetation of degraded sites is a prime goal of restoration to avoid establishment of invasive plants and to reduce soil erosion. We also evaluated whether the effects of the designed mixtures were consistent over time (represented by three harvests) which might reflect different environmental conditions. Here, we discuss the suppressive effects of the designed mixtures towards *J. aquatica* as controlled by the base communities, the diversified and constrained traits, and phylogenetic diversity (PD). While we observed that native community biomass strongly suppressed *J. aquatica* across different experimental periods, the manipulation of seed mass and leaf area resulted in later suppression, and PD only showed early effects on biomass of *J. aquatica*. This emphasizes that community development takes time, and that the long-term effects of the manipulated aspects could be more important.

Invasive biomass was always highest in the monoculture community, whereas it was lowest in the grass-dominated LfL reference mixture. One reason for such a pattern might be the fact that grasses are more competitive in the short term. Grasses produce more biomass in less time due to their extensive root system that allows for acquiring more nutrients in the soil upper layers [15,22]. Still, when the target is establishing a flower-rich community, other aspects rather than the percentage of grasses should be considered. In this study, we designed grassland communities by manipulating traits related to different plant strategies as an attempt to suppress *J. aquatica*. However, our approach could not yet identify which aspects should be directly manipulated in the community to increase the suppression effects on *J. aquatica*.

### 3.1. Home Community Effects on Invasive Biomass

The competition level experienced by invasive plants at home and non-home communities is determined by the niche space [16], while we excluded environmental variation in the greenhouse experiment. The designed wet grasslands, i.e., home communities of *J. aquatica*, did not completely suppress the native invader. We argue that the equally distributed invasive biomass in the home grassland indicates a well-established position of *J. aquatica* in such communities, therefore no limiting effects of niche pre-emption on invasive biomass can be expected [43]. Additionally, in home communities, the native species pool most likely provided stable relationships among species as, for example, root–root interactions decreasing invaders [44]. The environment exerted by home communities towards *J. aquatica* might operate via asymmetric competition [19].

Contrastingly, the non-home mesic grasslands presented uneven suppressive effects. Its species pool might have offered non-established or not-previously-experienced relationships among species, which caused the invader to perform better or worse in such communities when compared to the result in home communities. Furthermore, the native invader might have presented a greater neighbor tolerance in these communities in comparison to home communities [17]. This might have occurred within the non-home communities designed for this study, in which the interactions among species were not constant, and the more tolerant *J. aquatica* must have filled every possible niche, therefore varying in abundance or biomass. Notably, effects resulting from differences in soil moisture of the experimental communities were minimal, because all communities were under the influence of the same irrigation treatment. Thus, in non-home communities, niche pre-empting was more relevant for controlling invasion than limiting trait similarities [45]. Therefore, home communities show promising trends for the suppression of invader biomass when environmental conditions fit the community requirements [8].

### 3.2. Native Biomass and Traits Suppressing the Invasive Plant

No clear pattern was observed in the suppressive effects of constrained traits towards *J. aquatica*. This contradicts the expectation that communities with similar traits to those of *J. aquatica* would outcompete this plant via limiting similarity, as observed in previous studies [22]. Still, diversified traits (i.e., not limited by the trait values yielded by *J. aquatica*) can outcompete the invader through consistent and complementary native biomass production [11], which we also observed in this experiment. This means that diverse traits might represent broader response possibilities to the environment and are, therefore, more relevant for a given plant community than trait similarities [46]. However, we also need to consider that the effects of functional diversity or traits might be site- or context-specific [47]. Therefore, further studies should consider the effects of traits on invasion resistance under different environmental conditions or at different stages of community development [48]. This connects to the observation that high native biomass reduced *J. aquatica* biomass; this effect was already observed in previous studies [49]. Highly diverse communities are more productive than low diversity ones. However, native community biomass might be preferentially the result of plant traits expressed by individual species. The issue as to whether some plant traits are directly related to plant growth or if they are a cumulative result of secondary cross-correlations is still unresolved [50].

When testing those plant traits, we saw that leaf area and seed mass had significant effects on *J. aquatica* biomass, albeit only in the final period of the experiment. Increased seed mass correlated with increased invasive biomass, while increased leaf area correlated with decreased *J. aquatica* biomass. Small seeds are larger in number, quick to germinate but more sensitive to environmental stress [29], which was mostly absent in the greenhouse experiment. The suppressive effect therefore might be due to higher density of small-seeded species, that resulted in more effective suppression as also reported by other authors [51]. After an initial fast germination of small seeds, the leftover larger seeds with the same expected growth rates as *J. aquatica* could not outcompete the invasive species. When growing, increased leaf area is related to resource acquisition through photosynthetic activity [31], and promotes a competitive shading effect. This asymmetric competition for light can decrease shade-sensitive species like *J. aquatica*. High growth rate and high specific leaf area are key traits that determine competitive resistance of native communities against plant invasions [27,52], and can explain invasion success in already established plant communities.

### 3.3. Temporal Trends in the Suppression of J. aquatica

Because PD values did not depend on species abundance, they should be considered as a reliable measure (together with base community type) for explaining the patterns of native and invasive biomass observed in this study. Increased PD was previously shown to reduce invader abundance, but also depending on the effects of vegetation gaps and availability of bare soil [53]. Contrarily to what was observed in previous studies, lower values of PD decreased invasive plant biomass at the beginning of the experiment, but this effect did not last across the three experimental periods. This might be explained by the correlation between PD and trait similarity, indicating that closely related species show similar trait values [20], and compete strongly for similar resources [21]. However, the effect of PD changed after the initial period; with increasing PD, there was an initial positive effect on biomass of *J. aquatica*, possibly coinciding with the period in which nutrient availability was higher. This argument agrees with the observation that significant effects of PD are mostly occurring in early stages of community development [54]. Still, potential effects of nutrient limitation in our experiment were attenuated by the fertilization implemented during the second and third experimental periods. Finally, the relationship between plant invasion and PD might be mostly related to the actual impact of invasion towards the recipient community rather than the effects of the native community towards the invader [54]. Due to these controversial findings, PD might not be a suitable tool for suppressing invaders in the long term.

*J. aquatica* biomass changed across experimental periods, probably following trends in community development, whereas seasonal effects are less likely to occur in a greenhouse experiment. Invasive species were found to present mostly traits related to competition and to tolerating limited resources, and lack stress tolerance [55]. Increasing native biomass decreased *J. aquatica* biomass permanently, but with a disproportionate effect (Appendix A). Such disproportional biomass relation was also observed as a pattern mediated by the effects of invader arrival [45].

Despite the effect of the traits such as plant height and leaf area, this might not represent the actual realized trait values, because CWM values have been calculated using an estimation of individual species abundance. This calculation was based on species-specific cutting tolerance values, thus assuming that species with lower cutting tolerance would be less abundant across the different experimental periods, which were determined by cutting events. Still, cutting can be considered as a ‘levelling’ intervention because, by cutting all plants at the same level, we could observe initial trait effects for those plants which were able to regrow. However, a ‘trait maturation’ effect can be associated with community development, because species composition might have changed over time due to different cutting tolerance. Thus, plant trait models can indicate hidden or developing traits that will characterize plant responses [56]. We argue that trait maturation influences community suppressive effects, with traits becoming more relevant with prolonged experimental time and, consequently, at later stages of community development. In the beginning, this study observed competition effects among seedlings, while towards later stages and with no re-sowing, the plant community reached some maturation. Such observations should be considered when designing seed mixtures for the implementation of grassland communities in managed areas that can be under risk of continuous plant invasions.

Future studies should experimentally manipulate other traits and competition aspects of biomass, and home vs non-home communities over longer timespans and under field conditions. This would greatly improve the understanding of the role of temporal effects of trait maturation, competition, and community development on plant invasions.

## 4. Materials and Methods

### 4.1. Plant Species Selection and Design of Seed Mixtures

To understand community effects on suppression of *J. aquatica*, we designed twelve seed mixtures with varying numbers of grasses and forbs based on two native communities in C Europe, i.e., wet grasslands (alliance Calthion, [57]) where the species typically occurs, and mesic grasslands (Arrhenatherion, [58]) that might become invaded. To reduce environmental uncertainty, invasibility of the communities was tested under standardized greenhouse conditions against a monoculture of the invasive plant, and a commonly used grassland mixture, i.e., BQSM-D2 produced by the Bavarian Research Centre for Agriculture (LfL; see Appendix A for all grasslands mixtures).

To evaluate invasion resistance of wet and mesic grassland communities, 36 native species were chosen based on their occurrence in S Germany and commercial availability. The species represented 15 families within varying frequencies. Seeds of all species (except *J. aquatica*) were obtained from Rieger-Hofmann GmbH, Blaufelden, or Johann Krimmer GmbH, Pulling, Germany. To ensure the same conditions in seed quality which can be affected by the production process, the LfL mix was prepared with seeds of the same origin as for the seeds used for generating the other mixtures.

Values for plant height, leaf area, and seed mass were extracted using the TR8 package of R [59]. Design of the seed mixture was based on the method of Laughlin [60], that allows for species assemblages converging on selected trait values, while diversifying others. This method consists in the use of a nonlinear optimization algorithm calculated with the function *selectSpecies* to define the relative abundances of the species composing a given mixture that will maximize RaoQ (as a measure of functional diversity), while also controlling for specific linear constraints. Hence, with this function, one can select a certain plant trait as diversified, and others as constrained according to a pre-defined threshold, thus increasing the potential of limiting similarity between the targeted seed mixture and potential invaders.

For each trait constrained when creating our grassland mixtures, we used the threshold values determined by the respective trait value presented by *J. aquatica*. Therefore, when constraining a trait as, e.g., plant height for our grassland mixtures, all species composing such mixtures were expected to be of similar height as the native invasive *J. aquatica*. In turn, diversifying plant height resulted in a community with diverse plant heights. This approach was selected to test the increased competitive ability of the target mixtures. These procedures resulted in a species abundance matrix with the desired diversified and constrained trait values and corresponding species proportions. Our grassland mixtures therefore contained plant species with one trait constrained to *J. aquatica* trait levels and the other one diversified. This resulted in grassland mixtures that also had different phylogenetic diversity values, while seed mixtures were always composed by three species of grasses and five forbs. The LfL mix was composed of six species of grasses and two forbs.

### 4.2. Experimental Design and Implementation

The experiment was established on May 2019 within the Centre of Greenhouses and Laboratories Dürnast, Technical University of Munich, Germany (48°24′ N, 11°41′ E). Trays (size 48 cm × 33 cm × 6 cm, 0.158 cm^2^) were filled with Stender^®^ Propagation substrate (Appendix A). Each tray was sown with 3 g m^−2^ of seeds from the grassland target species and 1 g m^−2^ of *J. aquatica*, to simulate invasion of an initial grassland community. This meant that small-seeded species were sown at higher densities that large-seeded ones. To ensure a homogeneous sowing pattern, seeds were mixed with perlite, that is physical-chemical inert [61]. A germination test of the sampled *J. aquatica* seeds showed adequate rates at 10–20 °C (Appendix A).

The treatments or grassland mixtures were coded according to community type (M, mesic; W, wet), trait constrained (Ph, plant height; La, leaf area; Sm, seed mass) and *trait diversified* (using the same plant traits as previously; Appendix A), respectively. Combinations of the three constrained and the three diversified traits (e.g., diversifying Ph while constraining La or diversifying Sm while constraining Ph) resulted in six plant traits treatments per community type (plus a *J. aquatica* monoculture and a LfL reference community), which were then replicated six times. The 84 experimental trays were labelled according to the corresponding treatment and randomly distributed on a floodable table inside the greenhouse.

The grassland communities and *J. aquatica* successfully established in all trays. Mean native community biomass was highest in the LfL grassland mixture used as a reference (Appendix A). Each tray was re-randomized weekly to a random position to avoid any edge effects. The overall timespan of the greenhouse experiment was almost a year divided in three periods, with the first harvest after 10 weeks (Period 1), the second after another 15 weeks (Period 2) and the last after another 22 weeks (Period 3). In autumn and winter, the time periods were increased to compensate for decreasing day length, while otherwise the growth conditions in the greenhouse did not change much, and minimum temperatures were kept at 16–21 °C. Fertilizer was added in periods 2 and 3 through irrigation water (see Appendix A). During summer, temperatures did rise up to 43 °C, but due to a daily irrigation regime via flooding the experimental table, the soil of experimental trays was never completely dry (K. Möhrle, personal observation).

### 4.3. Data Collection and Indices Calculation

Plant cover was visually estimated across all grassland mixtures and experimental periods. Plant biomass was manually harvested 1–2 cm above the soil surface, collected in paper bags and dried at 65 °C for 120 h (Binder ED400). After drying, plant biomass of the native community was separated from the biomass of *J. aquatica,* and both were measured using a Kern 572 precision scale.

Germination rates of each species were obtained through germination tests and literature survey (Appendix A). Since biomass of the sown native species could not be separated due to logistic constraints, we used the species-specific cutting tolerance factor of Briemle and Ellenberg [62] to estimate species abundance after each cutting event. Cutting tolerance ranged from 1 (not cutting tolerant) to 9 (promoted by cutting); therefore, a tolerance value of 5 indicated species not affected by cutting. Thus, the cutting tolerance factor was multiplied with species abundance in the original seed mixtures (Appendix A). This procedure was used to obtain the community-weighted mean (CWM) trait values, thus incorporating the relative abundances of individual species. We calculated CWM values of plant height, leaf area and seed mass for each grassland community, using the *functcomp* function of package FD in R [63]. Finally, we calculated phylogenetic diversity (PD) using the *phylomatic* tool [64], and the phylogenetic information based on the megatree of Zanne [65]. After converting the obtained dataset for each treatment to the appropriate. *tre* file data format, phylogenetic diversity was calculated in R using package *ape* [66] resulting in the Faith’s Index [67] for overall PD.

### 4.4. Statistical Analysis

Biomass values were standardized by dividing it by the number of weeks of the respective period. Biomass of *J. aquatica* was considered as a response variable, while the biomass of the native community was an explanatory variable. Prior to further analysis, outliers of individual mixtures were replaced by mean values calculated for each mixture in each period after assessing average biomass of *J. aquatica* and the native community with a Grubbs test. This resulted in six outliers in the mixtures M-Ph*La* and M-Sm*La* in period 3. Afterwards, all continuous variables were rescaled by the function scale in R Studio (by dividing the chosen columns by their standard deviation) for the calculation of the linear models, due to the large scale span of the data (e.g., from mm^2^ to mg).

First, we tested the overall communities for significant differences using simple linear models with one explanatory variable (‘mixture’) and one covariate (‘period’), and post hoc Tukey HSD tests from the package *agricolae* [68].

Then, we fitted full linear mixed-effects models (package lme4 [69]) followed by a Wald Chi-square test, to assess the effects of community type as well as constrained and diversified traits (additive effects and correlating effects). The experimental trays and the three experimental sampling periods were considered as random factors in our models. Afterwards, we assessed the effects of the CWM trait values, phylogenetic diversity and native biomass by fitting a full linear mixed-effects model followed by a Wald chi-square test. The commercial LfL mixture and the *J. aquatica* monoculture were used as references.

## 5. Conclusions

Native plant biomass and trait values provide insight on the initial community assembly and on the effects of trait maturation towards a native invader. Although no clear suppressive patterns were observed, greater native community biomass showed a continuous and disproportional suppressive effect on the invasive native plant, whereas functional trait effects emerged over time. Therefore, the use of trait-based information in the design of grassland communities was only moderately successful in suppressing an invasive native species, most likely due to community complexity and temporal effects. Finally, our results provide an environmental-friendly approach to create flower-rich grasslands by including community-based mechanisms in its design. This will allow the establishment of functionally diverse plant communities in degraded areas.

## Figures and Tables

**Figure 1 plants-10-00775-f001:**
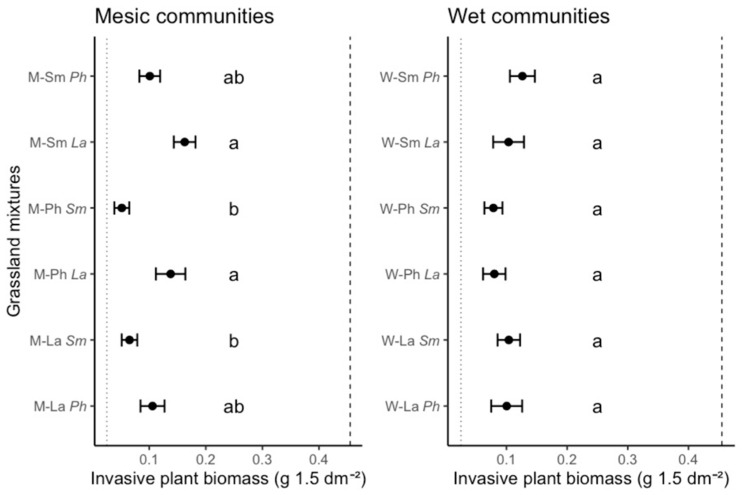
Grassland community effects on biomass of the native invasive *Jacobaea aquatica* (measured per tray area), in twelve trait-based grassland mixtures. Mesic (M) and wet communities (W), with standardized plant traits: Ph, plant height; La, leaf area; Sm, seed mass; first letters, constrained trait; and *second letters*, diversified trait. Vertical lines indicate as references the mean values of (i) a commercial grassland mixture (LfL, dotted line), and (ii) the *J. aquatica* monoculture (striped line). Significant differences were found among seed mixtures (χ^2^ = 39.3, df = 11, *p* < 0.001), but not between base communities (χ^2^ = 0.028, df = 1, *p* = 0.87, Appendix A); different letters indicate results of pairwise post-hoc Tukey tests (*p* < 0.05; mean ± SE; see Appendix A).

**Figure 2 plants-10-00775-f002:**
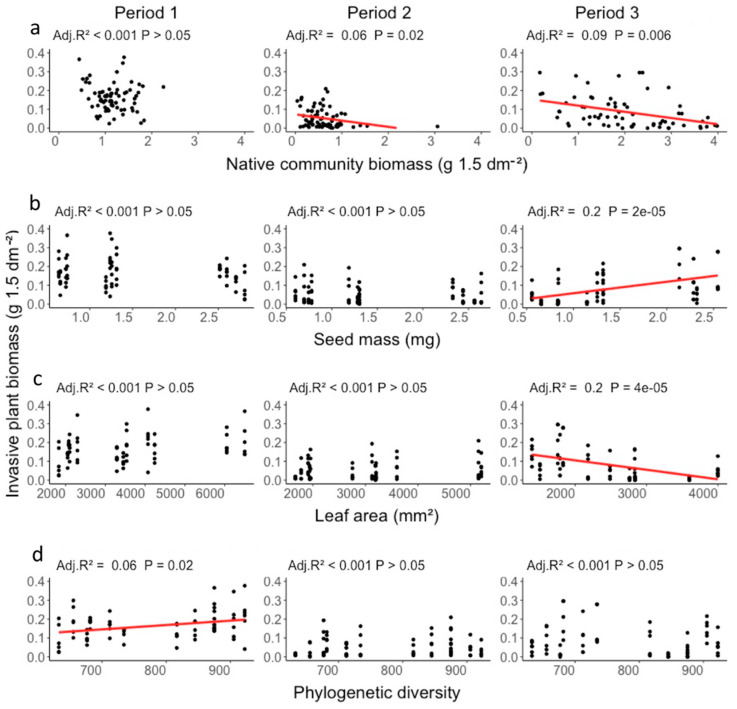
Effects of (**a**) native grassland biomass, (**b**) community-weighted mean seed mass and (**c**) leaf area, and (**d**) phylogenetic diversity on biomass of the invasive native *Jacobaea aquatica* (averages per week, biomass per tray) in three experimental periods (period 1, 10 weeks; period 2, 15 weeks; period 3, 22 weeks). Overall significant effects of native community biomass (χ^2^ = 16.0, Pr < 0.001), seed mass (χ^2^ = 10.9, df = 1, Pr < 0.001), leaf area (χ^2^ = 4.2, df = 1, Pr = 0.04) and phylogenetic diversity (χ^2^ = 5.3, df = 1, Pr = 0.02) are shown; regression lines were calculated using linear models.

**Table 1 plants-10-00775-t001:** Characteristics of the experimental grassland communities that control suppression of the invasive native *Jacobaea aquatica* (CWM, community-weighted means; biomass without the invasive plant; significant variables in bold). Results of a linear mixed effects model followed by a type III Wald Chi-square test with all variables, and experimental period included as random factor.

Variable	Chisq	Df	Pr (>χ^2^)
Native biomass	16.01	1	<0.0001
CWM Seed mass	10.95	1	0.0009
Phylogenetic diversity	5.35	1	0.0207
CWM Leaf area	4.21	1	0.0401
CWM Plant height	0.27	1	0.6044

## Data Availability

The data obtained in this study will be made available via media TUM.

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
