# Peer review of "Suppression of an Invasive Native Plant Species by Designed Grassland Communities"

_plants, 2021, doi:10.3390/plants10040775_

Round 1
Reviewer 1 Report
General comments to authors:
In this paper, the authors aimed to understand key drivers of invasion and engineer a grassland plant community that resists invasion by the native-invader Jacobea aquatica. Their communities are designed specifically to understand whether trait similarity (constrained traits) or diversity are better predictors of invasion resistance in two habitat types (mesic and wet). Overall, this paper is well-written and was enjoyable to read. The authors provided a good introduction to place their work in a proper context, their methods and statistical approach seem sound, and conclusions are warranted. I only have a few minor comments that the authors may want to consider to improve their manuscript. The most important comment, I believe, is that the authors should acknowledge the constraint of their study design- that they are essentially testing competition between seedlings, rather than invasion resistance of a community. To test the latter, they would stagger the introduction of the invader seeds for some time after establishment of the native seed mixes. I think they may have found different results in the importance of community traits, if they had followed that path.
Also, I believe the authors do not highlight a potentially important finding of theirs- that all communities, regardless of trait combinations, provided significant resistance to J. aquatica. This suggests that, although there is work to be done to understand functional traits and resistance to invasion, that having any community established (rather than disturbance) would provide resistance to invasion.
Specific comments to authors:
Abstract L15. Need another sentence set up the experiment, such as, "designed communities could be implemented by natural resource managers as a way to resist further invasion or reinvasion after control."
L58. This sentence is confusing, consider rewriting for clarity. I think the intended meaning is that invaders are less impacted by competitive interactions with co-occurring native species, but it is unclear because the terminology in the paper sometimes refers to “native invaders”. does this refer to co-occurring native species (one of which is invasive)?
L63. This paragraph lays the groundwork for community assembly and the role of niche similarity in structuring communities. But, the ideas need to be laid out a bit more clearly. There is some confusion in the way the authors refer to native and non-native invaders/alien, and home and non-home communities. It seems like they are arguing that normally co-occuring species will have reduced competitive interactions because they are co-adapted, but that species that do not normally co-occur will have stronger competitive interactions. This needs to be fleshed out more clearly.
L88. Perhaps use “resist” instead of “outcompete”?
L252. Also, the importance of trait similarity or diversity could be context or species-specific.
Conclusions. It seems that a logical next step would be to establish communities as done here, then after some period of time, introduce the invader. This would be more realistic, and provide a nice complimentary test of invasion resistance based on community traits.
Materials and Methods. The actual replication is unclear. Somewhere in this section list the number of replicates for each treatment combination (I think it is 4?). Also, it appears that the trays were grown, then aboveground biomass was harvest at intervals corresponding to the three periods. This is a bit unclear, but should be stated clearly that the same trays were used for measurements after harvesting (i.e., the sample periods are not independent)
L419. J. aquatica misspelled.
Author Response
Response to Reviewer 1
In this paper, the authors aimed to understand key drivers of invasion and engineer a grassland plant community that resists invasion by the native-invader Jacobaea aquatica. Their communities are designed specifically to understand whether trait similarity (constrained traits) or diversity are better predictors of invasion resistance in two habitat types (mesic and wet). Overall, this paper is well-written and was enjoyable to read. The authors provided a good introduction to place their work in a proper context, their methods and statistical approach seem sound, and conclusions are warranted. I only have a few minor comments that the authors may want to consider to improve their manuscript. The most important comment, I believe, is that the authors should acknowledge the constraint of their study design- that they are essentially testing competition between seedlings, rather than invasion resistance of a community. To test the latter, they would stagger the introduction of the invader seeds for some time after establishment of the native seed mixes. I think they may have found different results in the importance of community traits, if they had followed that path.
->We appreciate the positive comments of reviewer 1 regarding the design and development of our study, and we are glad that the reviewer found the manuscript enjoyable to read. We confirm that an early phase of community development was studied under controlled conditions. Of course, there may be slightly different patterns if the study would be conducted under field conditions or with already established plant communities. Nevertheless, since we monitored the experimental plant communities for over 10 months, we believe that 1–2 months after seeding (NB: seeding of plant communities was performed only once, at the beginning of the experiment), we cannot talk about ‘seedling competition’ any longer, but rather investigated plant competition during early community assembly. This aspect is now acknowledged in the revised version (l. 327–333). Finally, we believe that is important to point out that our experimental communities actually became very similar to the reference grasslands in S Germany in terms of species composition and biomass production.
Also, I believe the authors do not highlight a potentially important finding of theirs- that all communities, regardless of trait combinations, provided significant resistance to J. aquatica. This suggests that, although there is work to be done to understand functional traits and resistance to invasion, that having any community established (rather than disturbance) would provide resistance to invasion.
->Good point, that we unfortunately overlooked. Certainly, revegetation of degraded sites is a prime goal of restoration to avoid establishment of invasive alien plants and to reduce soil erosion. We have added that to the manuscript (l. 207–210).
Specific comments to authors:
Abstract L15. Need another sentence set up the experiment, such as, "designed communities could be implemented by natural resource managers as a way to resist further invasion or reinvasion after control."
->We expanded the abstract by the sentence “This study aimed at designing grassland communities, tested in a greenhouse experiment, to determine key drivers of initial J. aquaticasuppression and give insight to weed suppression without pesticides.” (l. 13–25)
L58. This sentence is confusing, consider rewriting for clarity. I think the intended meaning is that invaders are less impacted by competitive interactions with co-occurring native species, but it is unclear because the terminology in the paper sometimes refers to “native invaders”. does this refer to co-occurring native species (one of which is invasive)?
->We expanded this paragraph with “The suppressive effect of such invasive native species is, however, not higher than the one of other native species, but they are more tolerant and less impacted by competitive interactions with co-occurring natives [17].” (l. 58–60).
L63. This paragraph lays the groundwork for community assembly and the role of niche similarity in structuring communities. But, the ideas need to be laid out a bit more clearly. There is some confusion in the way the authors refer to native and non-native invaders/alien, and home and non-home communities. It seems like they are arguing that normally co-occurring species will have reduced competitive interactions because they are co-adapted, but that species that do not normally co-occur will have stronger competitive interactions. This needs to be fleshed out more clearly.
->We expanded this explanation by “Thus, we expect the non-home species to express stronger competitive interactions towards the invader, and the home community to have weaker competition, therefore enabling the invader to grow better.” (l. 60–63) and with the correction of the comment above.
L88. Perhaps use “resist” instead of “outcompete”?
->Changed to “resist” (l. 89)
L252. Also, the importance of trait similarity or diversity could be context or species-specific.
->Yes, we corrected to “This means that diverse traits might represent broader response possibilities to the environment and are, therefore, more relevant for a given plant community than trait similarities [46]. However, we also need to consider that the effects of functional diversity or traits might be site- or context-specific [47]. Therefore, further studies might consider the effects of traits on invasion resistance under different environmental conditions or at different stages of community development[48].” (l. 262–268)
Conclusions. It seems that a logical next step would be to establish communities as done here, then after some period of time, introduce the invader. This would be more realistic, and provide a nice complimentary test of invasion resistance based on community traits.
->We agree with the reviewer comment and inform that this research will be followed up by standardised experiments under field conditions.
Materials and Methods. The actual replication is unclear. Somewhere in this section list the number of replicates for each treatment combination (I think it is 4?). Also, it appears that the trays were grown, then aboveground biomass was harvest at intervals corresponding to the three periods. This is a bit unclear, but should be stated clearly that the same trays were used for measurements after harvesting (i.e., the sample periods are not independent)
->We corrected accordingly: “The treatments or grassland mixtures were coded according to community type (M, mesic; W, wet), trait constrained (Ph, plant height; La, leaf area; Sm, seed mass) and trait diversified (using the same plant traits as previously; Table S7), respectively. Combinations of the three constrained and the three diversified traits (e.g. diversifying Ph while constraining La or diversifying Sm while constraining Ph) resulted in six plant traits treatments per community type (plus a J. aquatica monoculture and a LfL reference community), which were then replicated six times. The 84 experimental trays were la-belled according to the corresponding treatment and randomly distributed on a floodable table inside the greenhouse.” (l. 401–409) and “The experimental trays and the three experimental sampling periods were considered as random factors in our models.” (l. 460-461)
L419. J. aquatica misspelled
->We have corrected the name of the model plant species used in our study.

Reviewer 2 Report
The evaluated manuscript fits the profile of the journal Plants and its results are interesting and original.
In the manuscript, the authors present the results of some experiments carried out to control the development of an “invasive” weed from the grasslands of Central Europe using the competitive capacity of different native grasslands communities. The conclusions, which are not very categorical, point to the convenience of using trait-based information in the design of grassland communities as a control tool for invasive plants in such grasslands.
In addition to a work with a good experimental design, with interesting discussion and conclusions, for me, the interest of the article lies basically on two aspects:
- The proposal of a weed control method that avoids the use of pesticides and has a low impact on the natural environment.
- The launch of an interesting issue for debate: Is it correct to apply the term invasive to native plants?
For these reasons, it seems to me that this work deserves to be published in the Journal Plants.
Nevertheless, there are also some aspects that I believe should be modified in order to improve the quality of the article.
- A little more length should be dedicated to describe, in the Introduction section, the debate: Is it correct to apply the term invasive to native plants? Prominent botanists and ecologists think that the term "invasions" should be limited to allochthonous species. And Jacobaea aquatica is a native species in the study area.
- Likewise, in the Introduction section, the negative effects of Jacobaea aquatica on native systems and on human activities should be highlighted. If J. aquatica is a native plant in the study area, it is a natural component of native ecosystems. So, it is convenient to justify why the elimination of a native species will be beneficial, in a plant conservation context.
- The information contained in lines 339-347, under subtitle 5.3.1 Invasive native species, should be included in the Introduction section, instead of in the Material and Methods section.
- The synonym of this species, Senecio aquaticus Hill should be included in the Material and Methods section, as a name that is still used in several European Floras.
- Section 5.3. is complicated to understand, it should be rewritten and organized in a more understandable way.
- In the same way, it should be explained more clearly what is indicated in lines: 376-377: “For each trait constrained when creating our grassland mixtures, we used the thresh-old values determined by the values of J. aquatica”.
- In lines 421-422 it is stated that: “Germination of each species was obtained through germination tests and literature research (Tables S1, S10). However, in table S10, 10 of the species that appear there (it seems a lot to me!), the bibliographic reference to which the Mean germination in % data correspond is not indicated. It is indicated: personnal communication or personnal observation. This makes it impossible to contrast the information provided. A bibliographic reference should be included to support the data provided, in accordance with the methodology chosen by the authors.
- In the results section, in lines 109-110, it says: “Germination of each species was estimated to be around 65% (SD ± 21%) based on 109 previous germination tests (Table S1)”. It would be more correct to write: “Germination of each species was estimated to be around 65% (SD ± 21%), based on bibliographical references”.
Author Response
Response to reviewer 2
In the manuscript, the authors present the results of some experiments carried out to control the development of an “invasive” weed from the grasslands of Central Europe using the competitive capacity of different native grasslands communities. The conclusions, which are not very categorical, point to the convenience of using trait-based information in the design of grassland communities as a control tool for invasive plants in such grasslands.
->We thank reviewer 2 for these comments and for highlighting the importance of using trait-based information to increase invasion resistance of designed grasslands.
In addition to a work with a good experimental design, with interesting discussion and conclusions, for me, the interest of the article lies basically on two aspects:
The proposal of a weed control method that avoids the use of pesticides and has a low impact on the natural environment.
->We applied this additional information on l. 45-48 “However, as the widespread use of herbicides is harmful to the environment, attempts of controlling invasive plants in managed grasslands should focus on using community-based mechanisms, e.g., specifically designed seed mixtures.“
The launch of an interesting issue for debate: Is it correct to apply the term invasive to native plants?
->Please, see comment below regarding the discussion of considering native plants as invasive.
For these reasons, it seems to me that this work deserves to be published in the Journal Plants. Nevertheless, there are also some aspects that I believe should be modified in order to improve the quality of the article.
A little more length should be dedicated to describe, in the Introduction section, the debate: Is it correct to apply the term invasive to native plants? Prominent botanists and ecologists think that the term "invasions" should be limited to allochthonous species. And Jacobaea aquatica is a native species in the study area.
->We provided more information to this topic by l. 101–104 “Although being native to wet grasslands, J. aquatica is considered in this study as ‘invasive species’ because of its population dynamics with rapid spread, overabundance, and potential impacts on farms [14].”
Likewise, in the Introduction section, the negative effects of Jacobaea aquatica on native systems and on human activities should be highlighted. If J. aquatica is a native plant in the study area, it is a natural component of native ecosystems. So, it is convenient to justify why the elimination of a native species will be beneficial, in a plant conservation context.
->We expanded the explanation for this issue in l. 90–101 as following “In this study, we tested the resistance of experimental grasslands to Jacobaea aquatica (synonym for Senecio aquaticus, Hill, Asteraceae), a native poisonous herb recently be-coming overabundant in wet grasslands of Central Europe [33]. Here, Jacobaea aquatica has become a major invasive native plant [33], [34]. This short-lived perennial contains pyrrolizidine alkaloids that are resistant to drying or ensiling, and therefore persistent in the fodder. This poses a threat to livestock and humans through transferring of the poisonous components within the food chain [35]. Establishment is most common in disturbed moist to wet grasslands with altered management, where the species responds to gaps with fast germination and high growth rates [32]. The plant forms rosettes, and large amounts of wind-dispersed seeds are produced in its second year [36]. It can flower already at low height and shortly after being cut [37], [38], but is sensitive to shading, as observed in low-intensity or abandoned grasslands [39] [M.T. Krieger, unpublished results].”
The information contained in lines 339-347, under subtitle 5.3.1 Invasive native species, should be included in the Introduction section, instead of in the Material and Methods section.
->As suggested by the reviewer, we moved this part to the introduction, l. 90–104.
The synonym of this species, Senecio aquaticus Hill should be included in the Material and Methods section, as a name that is still used in several European Floras.
->We added the requested synonym in l. 91
Section 5.3. is complicated to understand, it should be rewritten and organized in a more understandable way.
->We applied the requested changes also according to the comments of reviewer 1 in l. 392–409. Additionally, we have rewritten the part explaining the statistical analysis in order to make it more understandable (l. 446-464).
In the same way, it should be explained more clearly what is indicated in lines: 376-377: “For each trait constrained when creating our grassland mixtures, we used the thresh-old values determined by the values of J. aquatica”.
->We expanded the information in l. 378–382 to clarify this issue.
In lines 421-422 it is stated that: “Germination of each species was obtained through germination tests and literature research (Tables S1, S10). However, in table S10, 10 of the species that appear there (it seems a lot to me!), the bibliographic reference to which the Mean germination in % data correspond is not indicated. It is indicated: personnal communication or personnal observation. This makes it impossible to contrast the information provided. A bibliographic reference should be included to support the data provided, in accordance with the methodology chosen by the authors.
->To clarify, we changed this point to “Germination test performed by Kathrin Möhrle, Chair of Restauration Ecology”; “Germination test performed by Sandra Rojas Botero, Chair of Restauration Ecology personal communication” or “Germination test performed by Hugo E. Reyes Aldana, personnal communication” in the supplementary material (Table S10).
In the results section, in lines 109-110, it says: “Germination of each species was estimated to be around 65% (SD ± 21%) based on 109 previous germination tests (Table S1)”. It would be more correct to write: “Germination of each species was estimated to be around 65% (SD ± 21%), based on bibliographical references”.
->We also performed our own germination tests for some of the species used in the experiment, and changed this information in Table S1 according to the comment above. This information is also provided in the text (l. 124-126).
Additionally, we cross checked germination rates within the supplementary material (Table S1) and, in light of new information, changed the value of Trisetum flavescensto 47%, as well as added the appropriate reference in Table S10.
